# Polymer Dynamics in Glycerol–Water Mixtures

**DOI:** 10.3390/molecules28145506

**Published:** 2023-07-19

**Authors:** Janez Stepišnik

**Affiliations:** Faculty of Mathematics and Physics, University of Ljubljana, 1000 Ljubljana, Slovenia; janez.stepisnik@fmf.uni-lj.si; Tel.: +386-3131-7274

**Keywords:** molecular dynamics, glycerol–water mixture, Rouse model, NMR molecular self-diffusion measurement, biopolymer folding

## Abstract

Velocity correlation spectra (VAS) in binary mixtures of water and glycerol (G/W), obtained by measurements using the modulated gradient spin echo (MGSE) NMR method, were explained by the interactions of water molecules with clusters formed around the hydrophilic glycerol molecule, which drastically change the molecular dynamics and rheology of the mixture. It indicates a thickening of the shear viscosity, which could affect the dynamics of submerged macromolecules. The calculation of the polymer dynamics with the Langevin equations according to the Rouse model, where the friction was replaced by the memory function of the retarded friction, gave the dependence of the dynamics of the polymer on the rate of shear viscous properties of the solvent. The obtained formula was used to calculate the segmental VAS of the polymer when immersed in pure water and in a G/W mixture with 33 vol% glycerol content, taking into account the inverse proportionality between the solvent VAS and friction. The spectrum shows that in the G/W mixture, the fast movements of the polymer segments are strongly inhibited, which creates the conditions for slow processes caused by the internal interaction between the polymer segments, such as interactions that cause disordered polypeptides to spontaneously fold into biologically active protein molecules when immersed in such a solvent.

## 1. Introduction

A prerequisite for life is the ability of biopolymers such as DNA, RNA or proteins to fold from a disordered polypeptide into a unique structure of a biologically active protein molecule. Despite many research efforts, the folding mechanism is still poorly understood, especially in the case of biopolymers immersed in certain liquids or their suspensions. In the latter, as a solvent, it mainly refers to liquids with a hydrogen bond, such as water, alcohol, glycerol and their mixtures, among which a significant part of research is focused on glycerol–water mixtures (GW) [1]. In order to understand the role of molecular interactions in the polymer folding in these blends, various measurement methods were used: measurements of thermodynamic properties [2], broadband dielectric measurements [3,4], infrared [5] and Raman spectroscopy [6]. These studies reveal changes in molecular structure beyond the first-neighbor level, but they do not reveal the general properties of the mixture and their influence on molecular dynamics. Various NMR methods were also used to characterize the structural and kinetics of protein reorganization [7,8,9,10,11]. A very important contribution to the understanding of its dynamics was made by measurements of the molecular self-diffusion coefficient *D*, which was initially measured by using the tracking technique [12] and the interferometric method [13], but a particularly important contribution was made by NMR spin echo methods in an inhomogeneous magnetic field [14]. These methods use the magnetic field gradient ∇|B|=G (MFG) to detect the translation displacement of molecules via an uneven precession of their atomic nuclear spins. Particularly important are results on G/W mixtures [9,13,15] confirming the validity of the Arrhenius behavior and the Stockes–Einstein relationship between *D* and viscosity η in a wide range of temperatures.

In our study, the measurement of molecular dynamics in liquids with the NMR gradient spin-echo method called modulated gradient spin echo (MGSE) [16] is discused. This method uses a sequence of Carr–Purcell–Meiboom–Gill radio frequency (RF) pulses [17,18] together with an MFG to modulate the spin phase and thus stroboscopically probe the power spectrum of the velocity autocorrelation function (VAS) [16,19]. The method was named modulated gradient spin echo (MGSE). The velocity auto-correlation function is a quantity that contains information not only about *D* but also about the basic processes of molecular interactions during diffusion [20] and therefore provides a deep insight into the physics of molecular dynamics in liquids [21,22,23]. The same information about the VAF of liquids was obtained by the neutron scatterings and also by the light scattering, but the short time scale of these methods cannot extract information about its long-time properties, as it is the case with the slow protein folding. Thus, the current understanding of these processes comes from more than experimental results: they come from computer simulations as well [24,25]; however, these largely depend on the chosen models.

## 2. Results and Discussion

A correction to the Rouse model by taking into account the hydrodynamic interaction in liquids was introduced with the Zimm model of polymer dynamics [26], which relates the Rouse modes of motion over long distances. However, in this study, we assume that the hydrodynamic interactions are hidden in the memory of the delayed effect of friction, as considered by Equation (Equation 5), and they are then expressed as the friction power spectrum γ(ω).

The results of MGSE VAS measurements of pure water and G/W mixtures with different glycerol content, shown in Figure 1, confirm that with increasing frequency, the D(ω) of water molecules in pure water also increases, while it decreases in G/W mixtures with a glycerol content greater than 10 vol%. Given that VAS is inversely proportional to shear rate viscosity, we can conclude that in pure water, we have a rheological property called viscosity thinning, while in a G/W mixture, there is viscosity thickening, both of which determine the friction of the molecules in the liquid. It means that in water, the fast movement feels less resistance than slow motion, while it is the opposite in a G/W mixture. In the reference [27], the MGSE measurements of VAS in G/W mixtures are shown in more detail, where it is shown that in the mixture with 5 vol% glycerol content, there is no significant difference in the form of D(ω) than that of water: it is only slightly shifted toward higher values. Thickening of the shear viscosity rate occurs in mixtures with a higher glycerol content.

With the aim of understanding the dynamics of the polymer immersed in liquids, we replace in Equation (Equation 8) for $D_{c}\left(\omega \right)$ and $\tau_{c}\left(\omega \right)$ the friction γ with the inverse values of the curve of VAS obtained by fitting the data of MGSE measurements for water and for the G/W mixture with 33 vol% glycerol. In this way, we obtain quite different VAS values of polymer segments, as shown in Figure 2, which reflect the dynamics of polymer segments in different liquids. Thus, the VAS of the polymer immersed in pure water shows a slight increase with frequency, which means that the motion damping decreases with frequency. But in the case of polymer immersed in the G/W mixture, VAS has an unusual peak at a frequency around 50 HZ with a width of about 150 Hz, which is twice as high as the VAS value at higher frequencies. Its shape demonstrates that the fast movement of the polymer segments in the G/W mixture is twice as damped compared to the slow movements with a frequency in a narrow band around 50 Hz. The inhibition of fast movements of polymer segments creates conditions for slow processes caused by possible weak interactions between polymer segments, such as when dealing with the folding of a polypeptide into a biologically active nRNA.

## 3. MGSE Method

The exceptionality of the MGSE method to probe directly the low-frequency range of VAS was already tested by measurements in different systems. It was used to study the molecular dynamics in ordinary liquids [28], fluid motion in porous media [29,30,31], segmental dynamics in polymers [32], and also the VAS in G/W mixtures [27], which is also the subject of this article.

The measurements of G/W mixtures were performed on a 100 MHz NMR spectrometer equipped with Maxwell gradient coils to generate MFG in steps up to 5.7 T/m at most. We used a CPMG sequence together with a static MFG in such a way that the product of the period of the CPMG sequence *T* and the amplitude of the MFG *G* were constant when changing *T*, which is equal to the inverse value of the spin phase modulation frequency. The high magnetic field and RF set-up magnet allow the precise modulation of the spin phase in the interval from 50 to 3000 Hz, where the maximum MFG of our system determines the highest frequency range of measurements. The results of our MGSE measurements of G/W mixtures with different glycerol concentrations are shown in Figure 1.

## 4. Molecular Dynamics in Glycerol/Water Mixture

As in the case of MGSE measurements of ordinary liquids, where the obtained low-frequency dependence of VAS could be explained neither by the decay of the long-tailed VAF, which is supposed to be the result of diffusion in the vortices of hydrodynamic fluctuations [33], nor by the model of short-term collisions between molecules, which are commonly used in molecular diffusion described by the Langevin equation (LE), where the friction γ and the random force f(t) are sufficient to calculate the VAS of molecular diffusion [34], it turns out that also the results of MGSE measurement in G/W mixtures differ from expected ones. We tried to find an explanation using the Langevin equation model, where in addition to collisions, the force of interaction between molecules also plays a role.These interactions are usually described by the Lennard–Jones potential, but this is difficult to effectively include in the description even in the case of interactions between molecules in pure water let alone between molecules in more complex systems, such as interactions in the G/W mixture between water–glycerol and glycerol–glycerol molecules. There were also unsuccessful attempts to solve the problem with the various modifications of the Lennard–Johnson potential [35]. But, it turned out that the unexpected form of VAS obtained in ordinary liquids can be described by LEs, in which the interaction between colliding molecules is described by a simple harmonic potential [28]. This may be explained by the fact that the fast motion of molecules at high temperatures averages out the molecular interactions to such an extent that the description with the harmonic force is quite sufficient [36]. Thus, the interaction of the *i*-th molecule with the mass $M_{i}$, which interacts with the harmonic constant *k* with the *n* nearest neighbors, can be described by a set of coupled LEs as
(1)$$\begin{array}{c} M_{i}\frac{dx_{i}^{2}}{dt^{2}}-\gamma \frac{dx_{i}}{dt}+k\sum_{j\neq i}^{n}\left(x_{i}-x_{j}-a\left(t\right)\right)=f_{i}\left(t\right). \end{array}$$
Inertialterms in all LEs can be neglected if we are dealing with light particles diffusing in a medium with strong friction γ, thus obtaining the low-frequency part of VAS in the range ω<<γ/M in the form
(2)$$\begin{array}{c} \langle v_{\omega i}v_{\omega i}^{+}\rangle =D\left(\omega \right)=\frac{k_{B}T}{\gamma }\frac{n+\tau_{c}^{2}\omega^{2}}{n^{2}+\tau_{c}^{2}\omega^{2}}, \end{array}$$
if neglecting the fluctuation of the location of the potential well created between the coupled molecules, a(t). Here, $k_{B}$ is the Boltzmann constant, $v_{\omega i}=\int_{-\infty }^{\infty }\frac{dx_{i}}{dt}\;e^{i\omega t}dt$ and $\tau_{c}=\gamma /k$ are the correlation time. At zero frequency, the spectrum $D\left(0\right)=\frac{k_{B}T}{n\gamma }$ depends on the number of coupled molecules, while at high frequencies, it is equal to $D\left(\infty \right)=\frac{k_{B}T}{\gamma }$, which is the diffusion rate of molecules that escape intermolecular capture and is equal to the Einstein–Smoluchowski diffusion coefficient (ES) [37,38]. This formula provides a good fit with the results of MGSE measurements in the case of pure water, as shown in Figure 1, as well as with the results of MGSE measurements of other liquids presented in reference [28]. However, Equation (Equation 2) cannot be applied to G/W mixtures with higher glycerol concentrations, which are presented also in Figure 1.

According to the references [39,40,41,42,43], in the G/W mixtures, the bond between water molecules as well as between glycerol molecules is weaker than the bond between water and glycerol molecules. It leads to the formation of clusters around hydrophilic glycerol molecules, whose interaction with unbound water significantly changes the molecular dynamics. When calculating the dynamics in the G/W mixture, we can no longer neglect the inertial terms of LEs for clusters due to their large mass, and in addition, to simplify the calculation, we use a slightly exaggerated assumption that the friction, γ, is equal to both water molecules and clusters and also that the coupling constant between small and large particles is also equal to *k*. Calculating with two sets of equations for $n_{c}$ heavy clusters and $n_{w}$ light water molecules, we obtain VAS for water molecules as [27]
(3)$$\begin{array}{c} D_{w}\left(\omega \right)=\frac{k_{B}T}{\gamma n_{w}}\left(\frac{\left(n_{w}-1\right)\tau^{2}\omega^{2}}{{\left(n_{c}+n_{w}\right)}^{2}+\tau^{2}\omega^{2}}+\frac{n_{c}n_{w}+\tau^{2}\omega^{2}+{\left(n_{w}-\frac{\omega^{2}}{\omega_{o}^{2}}\right)}^{2}}{\left(2n_{c}+n_{w}\right)n_{c}+\tau^{2}\omega^{2}+\frac{n_{c}^{2}\omega^{2}}{\tau^{2}\omega_{o}^{4}}+{\left(n_{w}-\frac{\omega^{2}}{\omega_{o}^{2}}\right)}^{2}}\right), \end{array}$$
Despite the oversimplifications, the calculated Equation (Equation 3) can be fitted to the measured VAS for G/W mixtures with higher glycerol content, as shown in Figure 1 for G/W mixtures with 33 vol% glycerol, if we assume that a water molecule only interacts with two of the neighboring water molecules and one cluster, and also neglecting interactions between clusters.

**Figure 1 molecules-28-05506-f001:**
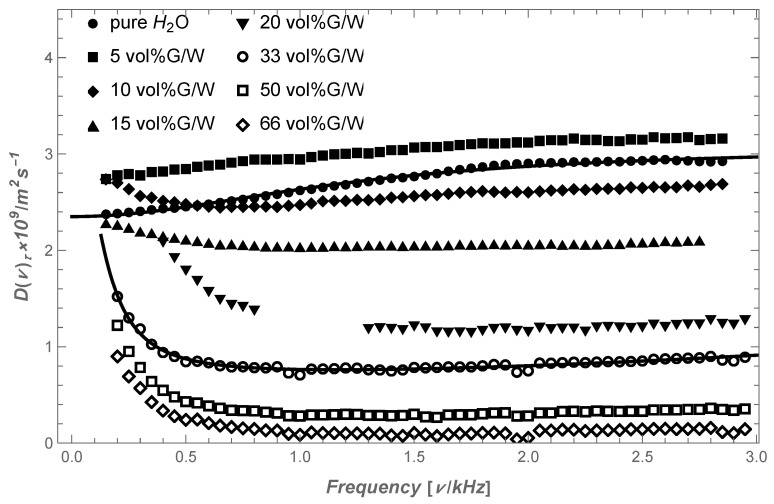
Velocity autocorelation spectra of water molecules in pure water and in G/W mixtures with different glycerol content. The curves represent the fitting with Equation (Equation 2) for water and with Equation (Equation 3) for G/W mixture.

The measurements showed that the self-diffusion coefficient in a G/W mixture decreases with the temperature decrease, while the viscosity increases according to the ES formula in a wider range around room temperature [44,45]. The SE formula was derived by assuming that friction is related to the fluid viscosity η as γ=sπRη, where a spherical object of radius *R* moves with small Reynolds numbers in the fluid according to the Navier–Stokes equations, where the factor *s* depends on the boundary condition. This indicates a direct relationship between the viscosity of the liquid and the diffusion constant. However, in the generalized LE [20,46], which is used in the next section in the calculation of polymer dynamics, hydrodynamic interactions and correlations between random forces at different times are taken into account by introducing friction forces with the memory kernel f(t), which means that the friction acting on the particle depends on the friction at an earlier time γ(t). Such a calculation leads to a generalized ES formula
(4)$$\begin{array}{c} D\left(\omega \right)=\frac{k_{B}T}{\gamma (\omega )}. \end{array}$$
Here, the friction spectrum is related to the shear viscosity velocity spectrum as γ(ω)=sπRη(ω). This formula was used to understand the swimming of microorganisms and sedimentation in liquids [47], where the viscoelastic modulus of a complex liquid influences the microscopic movement of small particles.

## 5. Dynamics of a Polymer in a Shear Rate Viscosity-Thickening Solvent

Experiments have shown [1,48] that the reconfiguration dynamics of unfolded proteins can be described sufficiently well by the standard Rouse (RC) spring chain model with internal friction, which was derived by Prince E. Rouse in 1953 [49,50]. The model consists of *N* balls of mass *m*, which are linearly connected by harmonic springs with spring constant *k* and move with a coefficient of friction γ. We will use this model in our case as well, but we will have to take into account that the conformational dynamics of the polymer are affected by the specific rheological properties of the solvent [51]. These properties could be obtained from MGSE measurements of molecular diffusion of G/W mixtures, considering the inverse proportionality between shear rate viscosity and VAS, $\eta \left(\omega \right)\approx D{\left(\omega \right)}^{-1}$, which reveals the viscosity-thickening shear rates in the G/W mixture at a glycerol content equal to or greater than 10 vol% [27]. Shear rate viscosity thickening might significantly affect the dynamics of a submerged polymer.

The Rouse model neglects all the details of the chain structure and is consequently only applicable to length scales at which the polymer appears as a very flexible chain with universal properties. Thus, the polymer is approximated by a linear sequence of Kuhn segments of length *b* large enough to permit the neglect of any stereochemical restriction of the orientation of the Kuhn segments relative to each other. The treatment of dynamics with LEs considers the polymer as a configuration of a linear sequence of Kuhn segments large enough to be described by the vector $R=(r_{1},r_{2},\cdots ,r_{m})$, where $r_{j}$ denotes the vector of displacement of the *j*th ball from its equilibrium position. In the overdamped limit, the chain dynamics can be described by coupled equations
(5)$$\begin{array}{c} k\left(r_{j-1}\left(t\right)+r_{j+1}\left(t\right)-2r_{j}\left(t\right)\right)-\int_{-\infty }^{0}\gamma \left(\tau \right)\frac{r_{j}\left(t+\tau \right)}{dt}d\tau +f_{j}\left(t\right)=0 \end{array}$$
where, unlike the standard procedure, we included a retarded effect of friction with the memory function as γ(t) [52]. In this approach, according to the oscillation and dissipation theorem, the friction is related to the random force $f_{j}\left(t\right)$, as
(6)$$\begin{array}{c} \langle f_{j}\left(t\right)f_{j}\left(0\right)\rangle =k_{B}T\gamma \left(t\right)\delta_{ij}\delta_{\alpha \beta }, \end{array}$$
where α and β represent the Cartesian coordinates of forces. In the further calculation, we use the usual procedure of converting Equation (Equation 5) to a set of Rouse modes [49], which allows the calculation of the segmental MSD as the sum of Rouse modes [53]. With the MGSE method, we only measure displacements in the direction of the applied MFG; therefore, with the Fourier transform of the one-dimensional mean-squared displacement, one-dimensional segmental VAS can be obtained, which can be approximated in the same way as in the reference [32] as
(7)$$\begin{array}{ccc} D{\left(\omega ,m\right)}_{R} & = & D_{c}\left(1+\sum_{p=1}^{m}\frac{\omega^{2}\tau_{c}{\left(\omega \right)}^{2}}{{\left(p/m\right)}^{4}+\omega^{2}\tau_{c}{\left(\omega \right)}^{2}}\right), \end{array}$$
where the number of modes can be approximately equal to the number of Kuhn segments, m≈N, in the case of a long chain. Here, $\tau_{c}\left(\omega \right)=\frac{b^{2}\gamma \left(\omega \right)}{3\pi^{2}k_{B}T}$ is the correlation time and $D_{c}=\frac{k_{B}T}{N\gamma (\omega )}$ is the center-of-mass diffusion coefficient of the coil [53]. If we convert the above summation over the modes in Equation (Equation 7) into a calculation with an integral [54], we obtain the VAS in the closed form as
(8)$$\begin{array}{ccc} D{\left(\omega \right)}_{R} & = & D_{c}\{1+\frac{1}{2}m\sqrt{\omega \tau_{c}\left(\omega \right)}[\frac{\pi }{\sqrt{2}}-\sqrt[{4}]{-1}(\tan^{-1}\left(\sqrt[{4}]{-\omega^{2}\tau_{c}{\left(\omega \right)}^{2}}\right)\\ & - & \tanh^{-1}\left(\sqrt[{4}]{-\omega^{2}\tau_{c}{\left(\omega \right)}^{2}}\right))]\}. \end{array}$$
where the dependence of the dynamics of the Rouse chain on the rheological characteristics of the solute is hidden in the friction spectrum in γ(ω) in $D_{c}\left(\omega \right)$ and in $\tau_{c}\left(\omega \right)$ [55].

**Figure 2 molecules-28-05506-f002:**
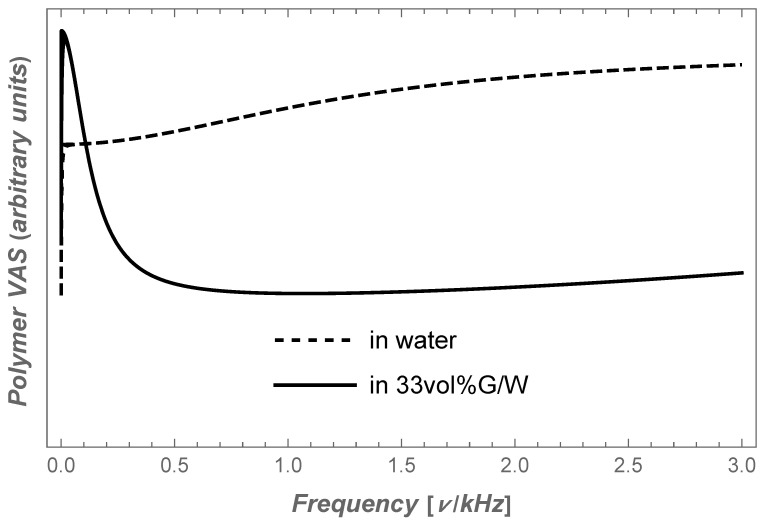
Autocorrelation velocity spectra of polymer immersed in water and in G/W blends with 33 vol% glycerol content (solid).

## 6. Conclusions

There have already been many attempts with various experimental methods and simulations to find an answer to the question about the influence of the diffusion and friction of a polymer on its folding, but the question remained unanswered [55,56]. Here, we try to contribute to these efforts by finding a link between the dynamics of the polymer and the rheological properties of the fluid in which the polymer is immersed. Measurements using the MGSE method in G/W mixtures showed an unusual change in the VAS of water at a sufficiently high glycerol content, which greatly changes the rheological properties of the liquid. The interactions of water with the hydroxyl clusters formed around the glycerol perturb the molecular dynamics in the G/W mixture to the extent that it changes from the thinning of the shear viscosity rate to thickening at a sufficiently high glycerol content. Calculation of the dynamics of a polymer immersed in G/W based on the Rouse model, where the fixed viscosity factor is replaced by a memory function that describes the retarded effect of friction, gives the VAS of the polymer, which confirms that thickening of the shear viscosity inhibits fast segmental motion to create conditions for slow molecular processes, such as folding polypeptides into a biologically active nRNA in such a solvent [57,58]. Figure 1 shows that this happens at a sufficient glycerol content when the rate of shear viscosity goes from thinning to thickening, somewhere in the range between 5 and 10 vol% of glycerol content.

## Data Availability

https://www.fmf.uni-lj.si/~stepisnik, accessed on 15 July 2023.

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
