# Peer review of "Polymer Dynamics in Glycerol–Water Mixtures"

_molecules, 2023, doi:10.3390/molecules28145506_

Round 1
Reviewer 1 Report
Please check the attached file.

Please check the attached file.
Author Response
Taking into account the comments, I slightly changed the introductory part, better described the experiment, deleted some references and replaced Fig.1 with one, which displays the results of a larger number of various G/W mixtures. From it is possible to estimate when the rheology of the mixture changes from one regime to another. I also corrected typos.
Reviewer 2 Report
This is a very interesting manuscript, presenting measurements of velocity autocorrelations of water molecules in water-glycerol mixtures. The resulting diffusion and friction data are then applied to model polymer dynamics in the mixed solvent. The most interesting finding is the prediction of a selective slowing down of fast polymer motions in the mixture relative to pure water. The results and analysis appear physically sound and should be of interest to a wide scientific audience. I recommend publication of the work after the author responds to the comments below:
1. The main physical argument for the unusual behavior of concentrated water-glycerol mixtures is the formation of water clusters sovalting the glycerol. It would be helpful for the readers if the author discussed the evidence for these clusters in more detail, especially if it is based on experimental data or on modeling.
2. There are several typographical errors in the text, such as
Abstract: Should be Langevin, not Lanevin
l.76 - should be Lennard-Jones, not Lennard-Johnson
Eq.5 - the integral is not correct
There are several typographical errors in the text, such as
Abstract: Should be Langevin, not Lanevin
l.76 - should be Lennard-Jones, not Lennard-Johnson
Eq.5 - the integral is not correct
Author Response
The formation of hydroxyl groups in mixtures of glycerol and water is confirmed in references [38-42], and are also confirmed by the form of water VAS obtained by our MGSE measurements, if we believe the derivation with the Langevin equations. Typos were corrected.
Reviewer 3 Report
Stepisnik reports on the VAS spectra of binary mixtures of water and glycerol obtained by MGSE NMR experiments. The paper is relatively well written, especially Introduction is comprehensive. The design of the experiment is reasonable and result can potentially bring new ideas about the influence of diffusion and friction of polymers on their folding. Nevertheless, the paper is rather technical and thus it will perhaps attract attention of a narrower audience. Also the ratio of Introduction and theory compared to results is unbalanced. Thus, I would shorten Introduction or extend Results. I am missing the Methods part describing the experiment completely. This needs to be added.
Nevertheless, I think the paper deserves to be published in Molecules after the Minor revision.
Author Response
The introductory part and the theoretical part were corrected , and the experimental part was added.
Round 2
Reviewer 1 Report
My first 2 major points were fulfilled by changing Fig. 1, while the third one was partially solved.
In the received version of the manuscript only the first two minor points were solved and the last 7 were not corrected/considered. The majority of them can be considered as typos.
Good English; only some typos spotted.
Author Response
I apologize to reviewer 1 for overlooking some of his comments. Because of my clumsiness, I lost the last part of his comments when I copied them.
Now,
- I changed the title of sections 2 and 3, and added the references to the papers of M. Doi and S. R. Edward and M. Rubinstein and R. Colby.
-Mean Square Displacement was written without abbreviation,
-I made correction on line 6, on line 119, and on line 135.
-I change the parenthesis in Eq.8. and corrected Ref. [54]